# A Study of Treatment of Industrial Acidic Wastewaters with Stainless Steel Slags Using Pilot Trials

**DOI:** 10.3390/ma14174806

**Published:** 2021-08-25

**Authors:** Mattia De Colle, Rahul Puthucode, Andrey Karasev, Pär G. Jönsson

**Affiliations:** Department of Materials Science and Engineering, School of Industrial Engineering and Management, KTH Royal Institute of Technology, SE-100 44 Stockholm, Sweden; rahulpu@kth.se (R.P.); karasev@kth.se (A.K.); parj@kth.se (P.G.J.)

**Keywords:** stainless steel slag, recycling, acidic wastewater treatment, upscale trials, mixing time

## Abstract

Different stainless steel slags have been successfully employed in previous experiments, for the treatment of industrial acidic wastewaters. Although, before this technology can be implemented on an industrial scale, upscaled pilot experiments need to be performed. In this study, the parameters of the upscale trials, such as the volume and mixing speeds, are firstly tested by dispersing a NaCl tracer in a water bath. Mixing time trials are used to maintain constant mixing conditions when the volumes are increased to 70, 80 and 90 L, compared to the 1 L laboratory trials. Subsequently, the parameters obtained are used in pH buffering trials, where stainless steel slags are used as reactants, replicating the methodology of previous studies. Compared to laboratory trials, the study found only a minor loss of efficiency. Specifically, in previous studies, 39 g/L of slag was needed to buffer the pH of the acidic wastewaters. To reach similar pH values within the same time span, upscaled trials found a ratio of 43 g/L and 44 g/L when 70 and 90 L are used, respectively. Therefore, when the kinetic conditions are controlled, the technology appears to be scalable to higher volumes. This is an important finding that hopefully promotes further investments in this technology.

## 1. Introduction

The mining and metal industry is considered to be a mature sector, in terms of technology development and R&D spending. In 2011, Filippou and King [1] estimated that less than 1% of the revenues of the whole sector were used to finance R&D projects. The authors claimed that this declining trend started around the 1980s. There are several compounding factors that hinder innovation in this sector. The most important one is the prohibitive startup costs that often require big capital investments. Opening new mines or metallurgical plants, as well as changing industrial processes or feedstock materials, are inevitably connected to big financial uncertainties. Therefore, a conversative thinking has usually been preferred. The result is a phenomenon called “technological lock-in”, which is defined as the tendency to maintain the status quo, due to several barriers impeding disruptive changes from happening [2].

Although, the mining and metals industry is already under pressure by several environmental constraints that require immediate actions. For example, the reduced quality of ores increases the percentage of waste generated [3]. Moreover, the rising carbon emissions pricing [4] is a serious threat to the profitability of the whole sector. Governments are tightening the restrictions around the generation of waste and the opening of new landfills, nudging companies for a change [5]. The public is also pushing for a greener and more sustainable production of goods, adding pressure to the companies to strive for an improved sustainability.

Metallurgical slags are among the most abundant by-products that are generated by the metals industry. Luckily, in many European countries, the technology to recycle those materials is mature enough to avoid excessive landfilling, mostly by employing slags as construction materials [6]. Although, in the case of stainless steel slags, such applications are not viable, due to the composition of the material, forcing their producers to dispose most of their by-products in landfills. In fact, the presence of high concentrations of heavy metals, such as chromium, impedes the use of these slags in such applications [7]. The production of slag in 2015, in Sweden alone, accounted for 1.35 Mton, of which 0.27 Mton came from stainless steel production, which is roughly 20%, as highlighted by the Swedish steel agency Jernkontoret in its latest report [8]. Although, the report shows that blast-furnace (BF) and basic oxygen furnace (BOF) slags are almost entirely valorized. The same applies for low-alloyed electrical arc furnace (L-EAF) slags. The most critical slags to valorize are high-alloyed electrical arc furnace (H-EAF) slags and argon oxygen decarburization (AOD) slags, which are both derived from processes that are associated with the production of stainless steel. By looking at the total of landfilled output, rather than the total amount produced, these slags constitute more than 70% of the total. It is evident that a solution for these kinds of materials is needed. Despite stainless steel being a niche production of a larger subset, it is responsible for most of the waste generation of the whole category.

The authors of this study have previously proposed a new application of stainless steel slags, which can contribute to the overall reduction in the landfilled output. Several stainless steel slags have been successfully tested as lime replacements for the treatment of industrial acidic wastewaters, generated in situ during the pickling process [9]. Lime is frequently used for the treatment of acidic wastewaters, but the reaction products are also often landfilled. Substituting slag with lime is projected to decrease the material input of stainless steel producers, thus decreasing the tons of by-products that are generated as a result. Moreover, if slag can be used as an effective reagent for the treatment of all kinds of industrial wastewaters, the sustainability of the whole stainless steel sector largely increases. 

Validating new technologies, especially for mature sectors, where the R&D budget is limited and a large capital is required, can be particularly difficult. Laboratory tests can only provide limited knowledge for the adoption of new technologies. Therefore, the aim of this study is to strengthen the technology validation, by replicating the same experiments that are conducted in laboratory settings in upscaled pilot trials. The primary focus of this study is to test whether the pH buffering of industrial acidic wastewaters, using stainless steel slag, can be replicated for bigger volumes than the one used in precedent laboratory trials [9,10]. Moreover, it is of interest to this study to determine the relationship between the volume of wastewaters to treat and the amount of slag needed to do so, when the kinetic conditions are controlled.

## 2. Materials and Methods

The first part of the study was to design a physical model that could control and replicate the same kinematic conditions when different volumes of wastewaters are tested. In fact, if the goal is to compare chemical reactions happening when using different volumes, the mixing conditions need to be the same to ensure a good comparison of the results. A physical model that relied on the dispersion of NaCl in a water bath was used to determine the relevant parameters to maintain a similar mixing performance across different volumes. Successively, the pH buffering trials were performed with the use of wastewaters and slags, utilizing the parameters determined by the physical model. The same methodology applied in precedent laboratory trials [9,10] was replicated to evaluate the amount of slag needed to buffer the pH to a value of approximately 9. The pH value of 9 was decided as a target to replicate the industrial processes, which use lime as a reactant to rise the pH level of the treated wastewaters.

### 2.1. Physical Model Design

How fast added slag can spread in the volume of wastewaters is one of the most important parameters to control when volumes are upscaled from laboratory conditions. When using small beakers (~1 L), the spreading of the slag can be considered to be instantaneous, given the small size of the container and the speed of the vortex generated by the magnetic stirrer. A homogenous spreading ensures that pH measurements performed at any given point of the volume are representative of the volume’s pH level. On the contrary, if the mixing phase is not instantaneous, until the slag is homogeneously distributed, it will react unevenly with the wastewaters across the whole volume. This means that the pH measured in any point during that phase of the upscale trial will likely not represent the pH value of the whole volume. This concern is the highest at the beginning of the trials, and it becomes less so the more the trials continue, since the slag has more time to spread in the volume. Thus, it is important to estimate how fast slag can homogenously distribute in the volume and try to minimize this time in comparison to when the first pH measurement occurs. In conclusion, to achieve kinematic conditions comparable across different volumes of wastewaters, an assessment of the homogenization speed of the solution needs to be performed.

To evaluate the time needed to reach a homogenous distribution of slag, the authors relied on a common method used in the field of metallurgy, which uses a physical water model to assess the kinematic conditions of AOD converters [11,12]. In particular, to determine the mixing performance of the stirring methods applied in agitated tanks, a parameter called “mixing time” has been extensively used. According to a literature survey conducted on the topic [13], there are two types of methods to determine the mixing time: the first is using one or multiple probes to measure local quantities (such as pH, conductivity or temperature). The second relies on global methods that are based on chemical reactions or optical analyses. Each method presents different challenges and limitations, so several sub-categories among these two types of models have been designed to circumvent some of them. However, no method imposed itself as the standard for this kind of investigation. In fact, for AOD converter simulations, several options have been used in the years. Wupperman et al. [14] used a photometer to detect the variation in color when a blue tracer is injected in a transparent tank filled with water. Moreover, Samuelsson et al. [15] used a local method of investigation instead, measuring the variation in pH when CO_2_ is injected in a tank filled with a NaOH-H_2_O solution. Other studies relied on models that measured the dispersion of a tracer (usually NaCl or KCl solutions) instead, in a tank filled with water [16,17,18,19,20]. The tracer changes the electric conductivity of pure water, thus conductivity probes can be used to evaluate its variation over time. This last method, specifically with using a NaCl tracer, was also used in this study.

To minimize the effect of local measurements, after the tracer is inserted, two probes placed at two different positions were used to measure the local conductivities of the solution over time. In accordance with the majority of the studies analyzed in the literature survey reported above, the water bath is considered to be homogenized when the conductivity of the probe C_i_ is equal to its final value C_final_ ± 5%. The last time when C_i_ falls outside the range C_final_ ± 5% is called the “mixing time” (hereafter denoted as T_m_). In other words, the “mixing time” is defined as the time after which the conductivity (C_i_) differs by no more than 5% of its final value. When two or multiple probes are used, T_m_ is defined as the greatest value between all the individual mixing times. A graphical example of how T_m_ can be determined from the conductivity measurement of a single probe is shown in Figure 1, where the ratio C_i_/C_final_ is plotted as a function of time. In the example, T_m_ (36 s) is calculated as being the last time when the curve exceeds the 1.00 ± 0.05 range.

#### Trial 1: Mixing Time Trials

As previously mentioned, the T_m_ value will be used to determine a set of parameters that will ensure that the same kinematic conditions are reached, when different volumes are tested. Moreover, if the T_m_ value is sufficiently smaller compared to the time when the first pH measurement occurs, the conditions in an upscale environment can be compared to the ones obtained in laboratory conditions. In previous experiments [9,10], a 1 L Erlenmeyer flask was used to treat the wastewaters with slag. It is assumed that for the low volumes and high mixing speeds selected in those trials, the kinematic conditions were shape-invariant. Therefore, a standard 200 L cylindrical plastic drum was chosen to contain the wastewaters. An overhead electric engine, paired with a stainless steel 3-blade impeller, was chosen as the stirring mechanism. The engine was secured on top of a steel frame, which also surrounded the drum.

The engine provided three rotational speeds of 225, 200 and 175 rpm, while the volumes chosen for the upscaled trials were 70, 80 and 90 L. The following several factors restricted the choice of the volumes: first, as the container remains the same, the ratio between height and radius of each volume changes. In addition, the position of the impeller relative to the ground was fixed too, meaning that its distance from the top of the volume increased with the increase in the volume itself. These two factors combined caused the vortex created by the engine to be different for each volume. Therefore, such differences needed to be minimized to maintain comparable T_m_ values for different volumes. Another restriction was set by the engine torque. In fact, the bigger the volume of water, the harder it was for the engine to spin the mass of liquid. This could result in excessive overheating and potential long-term failures.

Another important factor to consider was that the method chosen to evaluate the T_m_ value relies vastly on the position of the conductivity probes. Therefore, several positions were tested to have a proper assessment of the parameter. Since the drum can be approximated to a cylinder, thanks to its symmetrical properties there are only a limited number of combinations to test in order to cover the entirety of the volume. By assuming that the points that are harder to reach for the tracer are the ones at the borders of the volume, rather than at the center, 5 points were selected to test different probe positions. The points chosen are shown in Figure 2. Different combinations of rotational speeds of the engine, volumes of water and probe positions were tested. For almost all combinations, triplicate measurements were made to ensure good replicability. In total 30 trials were performed, and they were grouped according to Table 1.

The primary focus of the investigation was understanding whether the rotational speed needed to be adjusted per volume, or if the T_m_ value was speed-invariant. In this study the rotational speed was considered to be the most important factor in changing the kinematic conditions of the trial, so it was the parameter for which most of the measurements were carried out. For those trials, the two probes were positioned in an A & B configuration, as observed in Figure 2. The bottom of the drum, especially in its peripheral positions, was assumed to be the hardest part to reach by the added tracer. Therefore, the A & B and C & D configurations were preferred. In fact, it was assumed that the T_m_ values measured when the probes were placed at the bottom of the barrel would be higher than the ones obtained if the probes were placed on top of the volume of water. To test this hypothesis, the probes were also positioned in an A & E configuration. 

For each trial, a 20 mL of 20 wt% NaCl solution was used as a tracer. The trial started by activating the engine. Thereafter, the conductivity of the water bath was measured throughout the experiment. Specifically, at t = 0 s the tracer was inserted, and the change in conductivity was measured by the two probes every second. The trials stopped once both conductivity measures plateaued on a stable value. The T_m_ value was then calculated for each probe individually, and the value for the trial was selected as the highest between the two measured values.

### 2.2. pH Buffering Trials

The final goal of this study was to compare the results of the upscaled trials with previous laboratory experiments, to check whether the use of the same metallurgical slags as reactants for the treatment of acidic wastewaters can be replicated in an upscaled environment. Moreover, the trials aim at finding the relationship between the volume of treated wastewaters and the amount of slag being used, when the mixing conditions are kept constant. Therefore, the pH buffering methods chosen for the wastewater treatment and the material properties were kept aligned with those used in previous trials, to ensure a proper comparison between them.

#### 2.2.1. Sample Preparation

Given the large amount of wastewater needed per trial, the industrial upscale experiment was conducted in situ at Outokumpu Stainless, in Avesta (SWE). Therefore, compared to previous studies performed on four different slags, only two where available on site, but only one (namely, slag type “O1”) did not require being crushed [9]. Although, before using the material as retrieved from the slag yard, some operations were still needed. First, the slag provided was quite wet since it is usually water cooled when it is being disposed. Moreover, it is also stored in an outdoors environment. Slag wetness is not a problem intrinsically. However, to have a reliable estimation of the weights used, the experiments require a dry sample. Additionally, there were several impurities present in the slag, such as gravel or residual rock pieces, so sieving the material was required to remove the impurities. Therefore, the material was dried at 105 °C for 24 h and later sieved through a mesh of 350 µm for 15 min. This did not precisely replicate the conditions used in previous studies (which used a mesh of 1 mm). However, according to previous particle size analysis conducted on the same slag [9], 87% of the volume of the slag was made of particles smaller than 350 µm, ensuring, in principle, a good comparison between the two powders being used.

#### 2.2.2. Trial 2: Replication of Stepwise and Single-Step Dosing Methodology

Despite the constraints posed by the new environment, to effectively compare the results from the current upscale trial to the ones obtained by previous studies [9,10], the same methodology needs to be replicated too. Therefore, the “step-wise dosing” and “single-step dosing” methodologies developed previously, are used once again in this study. Since the single-dosing method operates by trial and error, by adjusting the quantity of reactant necessary to perform the trials, it usually requires a lot of attempts before the optimal quantity to reach pH 9.0 ± 0.2 is found. Therefore, a preliminary stepwise dosing methodology was used to narrow the range of investigation and trial numbers. In previous experiments [9,10], this method has been useful in correctly identifying the order of magnitude of the slag mass to use, avoiding unnecessary waste of materials during the single-step dosing method. Nine trials have been performed during this part of the study and their characteristics are listed in Table 2.

The experimental procedure for the stepwise dosing trials was as follows:At t = 0 a very small amount of reactant of weight w_1_ was added to the wastewaters. The reactant was inserted at the center of the vortex to promote good mixing.At t = 10 min and t = 20 min the pH value of the volume was measured.At t = 30 min the pH value was measured again, if |pH_30_ − pH_20_| ≥ 0.3 the pH was measured again at intervals of 10 min until |pH_i+10min_ − pH_i_| ≤ 0.3. Once the previous condition was met, if the pH value was not equal to 9.0 ± 0.2 the procedure was repeated from step 1 adding a new weight w_2_.When the pH value was 9.0 ± 0.2, the trial was stopped. The total amount of reactant w_tot_ was calculated, by summing all the weights used in the various cycles performed to reach the final pH value.

A schematic representation of the pH evolution over time of a stepwise dosing trial is shown in Figure 3.

In the case of single-step dosing trials, the procedure was also replicated from previous studies [9,10], changing only the times when the pH values were measured. A schematic representation of the pH evolution over time of a single-step dosing trial is shown in Figure 4. The experimental procedure for the single-step dosing trials was as follows:By using the quantities measured during the stepwise dosing trials as a benchmark, an appropriate amount of reactant of known weight was mixed with the wastewaters at t = 0 min. The reactant was inserted at the center of the vortex to promote good mixing.The pH value was measured at t = 10, 20, 30, 40, 50, and 60 min.If the pH reached the value of 9.0 ± 0.2 at the 30 min mark, the trial was considered to be successful.

For both stepwise dosing and single-step dosing methods, all pH measurements were performed by extracting a wastewater sample of approximately 20 mL from the surface of the agitated tank, close to the vortex generated by the impeller. The pH value of the sample was registered when a stable measurement could be obtained. Afterwards, the liquid inside the sample was poured back into the agitated tank.

## 3. Results and Discussion

### 3.1. Trial 1: Mixing Time Trials

The results of the first set of 18 trials, with the probes in the A & B configuration, are compounded in Table 3. Per combination of rotational speed and volume of water used, triplicate measurement trials were produced. The T_m_ values of each trial were averaged together, to provide a more statistically significant measure. In the first nine trials, volumes of 90, 80, and 70 L were stirred at 225 rpm. These trials were conducted to test how much changing the volume could influence the T_m_ value, when the stirring speed remains unchanged. The average T_m_ value slightly decreased with the decrease in the volume, from 10.3 s, when 90 L were used, to 9.7 s when 80 L were used, and finally 9.3 s when 70 L were used. The variation is in the range of 1 s.

To test the relationship between different volumes of water and rotational speeds even further, nine other trials were conducted. Here, the tank was filled with 80 L of water and the volume was stirred three more times at 200 rpm, while the volumes of 90 L and 70 L were also stirred three times at 175 rpm. Compared to the 225 rpm trials, when a volume of 80 L was stirred at 200 rpm, there was a slight increase in the T_m_ value, with an average value of 10.3 s. The same average T_m_ value of 10.3 s can be found in the 70 L triplicates that were stirred at 175 rpm. At first glance, it might seem appropriate to decrease the rotational speed to accommodate for the decreased volume. However, the same average T_m_ value of 10.3 s was also found when a volume of 90 L was stirred at 175 rpm. The most plausible interpretation is that the precision of the measurements is not sufficient to detect a meaningful variation in the T_m_ values, when parameters such as volume and rotational speed are changed in the ranges chosen for this study.

Twelve additional trials were performed, to test whether different probe positions could alter the T_m_ values. The results are compounded in Table 4. A water volume of 90 L was stirred at 225 rpm and the probes were positioned at the bottom, at a 90° angle, compared to the configuration A & B, namely, the positions that are indicated as C & D in Figure 2. Once again, the average T_m_ value was comparable to previous results. The variations detected were attributed to the imprecision of the test, rather than being correlated with the probe positions. The same can be said when the trials are performed using the probe configuration C & D, applied to volumes of 80 L and 70 L, stirred a 200 and 175 rpm, respectively.

Three more trials were performed with one probe located at the bottom of the tank, in position A, and one located in radial symmetry to it, but on the top of the tank, in position E. These trials were performed to test whether the probe at the bottom of the tank could measure a faster T_m_ value than the one at the top. In those trials, the contrary happened: in the three trials that were tested, the T_m_ values when the probes were positioned in A & E of 10 s, 8 s, and 7 s were measured. Therefore, when two probes are positioned at the bottom of the tank, either one of the two takes more time to reach its final conductivity value, compared to when one is positioned at the bottom and one at the top.

To summarize, a certain imprecision in the obtained results was expected to appear. This was also indicated in a study by Wupperman et al. [14], which detected a lower accuracy of local measurements compared to global ones. Although, for the scope of this study, the precision obtained during these trials is more than enough to ensure similar kinematic conditions when using different volumes. In fact, it is important to consider the context in which this physical model has been used. The experiments were designed to evaluate the time needed to achieve a homogenous spreading of the NaCl tracer in water, as a proxy for the spreading of slag. Although, the physical model is not very representative of what happens during the pH buffering trials. The ratio between the slag and wastewaters is greater than the one between the tracer and water. Moreover, slag does not fully dissolve as salt does, and it remains in a suspension in the liquid phase. Although, the time needed to homogenize the water bath after the NaCl injection, with the parameters chosen, is an order of magnitude lower compared to the time when the first pH measurement occurs. Thus, it is safe to assume that by the time the first pH measurement occurs, the slag is also homogenously distributed in the wastewaters. Moreover, a variation in the mixing time, between 9 and 10 s, can be considered to be negligible.

### 3.2. Trial 2: Replication of Stepwise and Single-Step Dosing Methodology

The pH buffering trials were separated in two different investigation methods. As for previous studies [9,10], the stepwise dosing method was used to narrow the range of reactant needed to buffer the pH to a value of 9.0 ± 0.2. The results are collected in Table 5. All the trials successfully reached the target of pH 9.0 ± 0.2. The quantity of slag needed for the stepwise dosing was 3 kg when a volume of 90 L was tested, and 2 kg when the volume was 70 L. Although, it is experimentally proven that the quantity of slag needed in the stepwise dosing method is always lower than the single-step dosing method [9,10]. This is because the latter method sets a time limit for the chemical reactions to happen (≤30 min, in accordance with the industrial requirements), which influences the amount of reactant to buffer the pH value to the desired target. Thus, a higher amount of slag is needed to reach the target pH value in 30 min, compared to the same quantity that can dissolve in a longer amount of time. This phenomenon did not happen when lime was used, which shows that lime has a higher reactivity than slag. In the case of lime, the quantity that was measured during the stepwise dosing was roughly the same as single-step dosing, due to how fast the lime powders reacted with the wastewaters [9].

Given the results from the stepwise dosing tests, 4000 g of slag was chosen as the starting quantity for the first single-step dosing trial, performed with 90 L of wastewaters (44.4 g/L). The target for the trial (IV-1) was to obtain a pH value of 9.0 ± 0.2 at the 30 min mark. The trial IV-1 succeeded in reaching pH_30min_ = 8.8. Thus, the same quantity was used again, to replicate the results and guarantee better reliability of the measurements. The second trial (IV-2) was also successful in reaching pH_30min_ = 9.1, but the third trial (IV-3) missed the mark, reaching only pH_30min_ = 7.7. The pH measurements of all the trials are shown in Figure 5a. It is interesting to point out that, despite the pH_30min_ value for the third trial widely differs from the other two, the pH_60min_ value is much more aligned. In fact, while trials IV-1 and IV-2 quickly plateau to a pH value of approximately nine, trial IV-3 reaches the same value with a slower rate. The same exact situation was obtained during the trials V-1, V-2, and V-3, which were performed with 70 L of wastewater and 3000 g of slag (43 g/L). In fact, trials V-1 and V-2 successfully hit the target value at the 30th minute, while trial V-3 lagged to reach the final pH value. Trial V-3 reached pH_30min_ = 7.0, compared to 9.2 and 9.0 of the two precedent trials, as shown in Figure 5b. A possible explanation to this phenomenon can be found in the variation in the wastewater composition. In fact, the wastewaters were extracted each time with a pump from the continuous flow of the industrial processes. Hence, the composition was not controlled during the trials. In both cases, using volumes of 90 L and 70 L, the first and second trials were conducted consecutively in the morning, whereas the third trials were conducted in the afternoon of the same day. If the wastewaters composition fluctuates between the morning and the afternoon trials, this can be an explanation for the differences in the pH buffering capacity of the slag being used. In fact, a different acid composition in the wastewaters might influence the dissolution rate of the slag. Therefore, an acidic environment that favors the slag dissolution results in a faster reaction. On the contrary, if the slag dissolution is slower, due to the different acidic content, the slag will take longer to dissolve and react with the acids. A confirmation of this hypothesis can be found by looking at the fact that the initial pH values are roughly the same in all the trials (between 1.49 and 1.57), and so are the final pH values, once the situation is closer to the reaction equilibrium. In fact, when a volume of 90 L is treated, the pH values are 9.1 (IV-1), 9.4 (IV-2), and 8.8 (IV-3), at minute 60. At minute 20, instead the values are 8.3, 8.9, and 6.8 for the respective trials. When a volume of 70 L is treated, at minute 60, the pH values are 9.5 (V-1), 9.3 (V-2), and 8.5 (V-3). At minute 20, the pH values are 8.9, 8.7, and 6.2 for the respective trials.

The results of these trials were also compared to similar experiments that were conducted in the past [9,10] to analyze both the amount of slag being used and their corresponding pH curves. In Figure 5c,d, two set of trials are presented that were conducted with the same kind of slag, extracted from the slag yard of the same steel-making factory. Compared to the pH curves that were obtained when using coarser slags, the current trials seem to reach the final pH value faster. The lack of pH measurements at 2 and 5 min makes it hard to draw a stronger conclusion, but the trends obtained by both trials, when wastewater volumes of 90 L and 70 L were tested, seem to resemble the one obtained when using finer slags (d) more. In fact, those trials are characterized by a fast rise in the pH values, followed by a long plateau, even though the quantity of slag used per liter is almost halved compared to the upscale ones. In addition, according to the results shown in Figure 5c, the quantity of slag is slightly higher, compared to the previous results shown in Figure 5d [10], which suggested that a reduced particle size also reduced the amount of slag needed to reduce the wastewaters. Although, the composition of the slag is variable and the effect of different minerals, regarding its capacity to buffer the pH values of the wastewaters, is still unknown. Similarly, a different composition of the wastewaters can alter the quantity needed to buffer its pH value. The different particle size is also a key factor that is hard to account for without being able to carry out proper measurements. Nonetheless, when only comparing the 90 L and 70 L trials to each other, the amount of slag needed to buffer the pH to a value of 9.0 ± 0.2 is roughly the same, while the pH levels reached are quite comparable, both at the 30 min and 60 min marks. Evidence of this can be observed in Figure 6, where a linear regression has been calculated with the amount of slag employed during the trials, with 90 and 70 L. The amount of slag from precedent trials has been added to the graph and compared to the calculated linear regression. As it is possible to notice in the enlarged area, the amount of slag that was measured with particle size <1 mm, falls very close to the regression line. In fact, the calculated amount of slag to buffer the pH of 1 L of wastewater, using the linear regression equation, should be 43.8 g. From empirical evidence, we know it to be 39 g [9], while the amount that is correspondent to the trial with a particle size <63 µm is 25 g [10]. In conclusion, the upscale trials proved to be successful in buffering the pH of bigger volumes of wastewaters, despite a different setup compared to the one used in the laboratory trials. When the particle size, composition of both the wastewaters and slag, as well as the kinematic conditions are controlled, the experiments seem to suggest that there is a linear relationship between the volume of wastewaters and the amount of slag needed to buffer their pH value.

The weight of slag needed to buffer the pH of the wastewaters with this setup is likely still too high for its successful use in an industrial implementation. Nonetheless, it is observed, from the laboratory trials, that the quantity per liter can be reduced by almost half, by decreasing the particle size distribution of the powders used [10]. During the current experiments, it was impossible to decrease the particle size of such large quantities of slags to replicate these findings. Although, given the comparable results between the pilot-scale trials and the first laboratory trials [9], it is expected that for upscaled volumes of wastewaters also, reducing the particle size of the powders would optimize the amount of slag needed. Another important factor to consider is that the pilot-scale trials have been conducted as batch tests, whereas the industrial case is a continuous flow process. Therefore, the optimal weight of the slag per liter of wastewaters, obtained during these trials, may not be indicative of the amount of slag needed in an industrial setting.

Finally, a consideration about the environmental aspect is due, given the nature of this study. Utilizing slag as a substitute for lime in the pH buffering of the wastewaters within the steel industry itself, could spare raw materials from entering the manufacturing process. Depending on the production process and internal reuse of generated by-products, this should constitute in a reduction in the landfilled output. In some cases, depending on the wastewater treatment process, spent lime is landfilled along with slag. Therefore, in that case, the use of slag constitutes a reduction in waste, since the lime is no longer needed. Sometimes, instead, the spent lime is recycled internally and used as a flux agent, thus generating more uncertainties in the possibility of substituting the material. In any case, the reduction in landfilled waste seems modest, as the volumes of spent lime, compared to the slag ones, are quite small. Also, the increase in reactant weight might be incompatible with the current wastewater treatment process, modifying the costs and composition of the landfilled output.

The use of slag as an acidic water treatment agent can be expanded to other industrial processes, translating in a larger volume of substituted lime. Along with the reduction in waste, the reduced need of lime translates in a reduction in CO_2_, since the material is produced through the calcination of limestone. Thus, the more lime substituted, the less carbon dioxide produced. Moreover, the composition of the solid residues after the pH buffering highly depends on the treated acid composition and the composition of the slag used for the pH buffering. Therefore, it is quite hard to predict what use can be conducted with the spent slag and how those residues can be further valorized for other uses. Additionally, the toxicity levels of the treated wastewaters were not analyzed during this study, since the validation of the slag used in this regard was already conducted in a more thorough experiment in a precedent study [9].

## 4. Conclusions

The aim of this study was to test an upscaled environment for the treatment of industrial acidic wastewaters with slag, and to compare the results with previous results from laboratory experiments. Moreover, since the aim was to compare the results between the 70 L and 90 L trials, the kinematic conditions were maintained constant. Specifically, a physical water model was used to determine the kinematic conditions, to find a set of parameters that could ensure similar mixing performances across the two different volumes. The study found that for almost all the combinations of rotational speeds of the impeller, the volumes tested, and the probe positions, the mixing times were 10 ± 2 s.

The results from the stepwise and single-step dosing methods of addition of the slag, developed in previous studies, have been replicated for the current experiments. The results showed that 44 g/L of slag and 43 g/L were needed to reach pH values of 9.0 ± 0.2 in 30 min, when 90 L and 70 L of wastewaters were tested. With the same method of slag addition, previous results estimated that 39 g/L (grain size <1 mm) and 25 g/L (grain size <63 µm) were the adequate quantities. A linear regression, calculated with the data collected during the upscaled trials, predicted that the quantity that is necessary to buffer the pH value to 9.0 ± 0.2 should be approximately 43.8 g/L.

In conclusion, the results of this study deepen the knowledge regarding the use slag for the pH buffering of the acidic wastewaters derived by the pickling process. More specifically, it provides reliable results to show that the material can provide an adequate treatment of the wastewaters, even when their testing volume is increased by 90 times, albeit when the mixing conditions are kept constant. Also, a relationship between the amount of slag and the liters of acidic wastewater is found. Although, the relationship is highly dependent on the properties and compositions of the slags used, as well as the kinetic conditions of the experiments.

## Figures and Tables

**Figure 1 materials-14-04806-f001:**
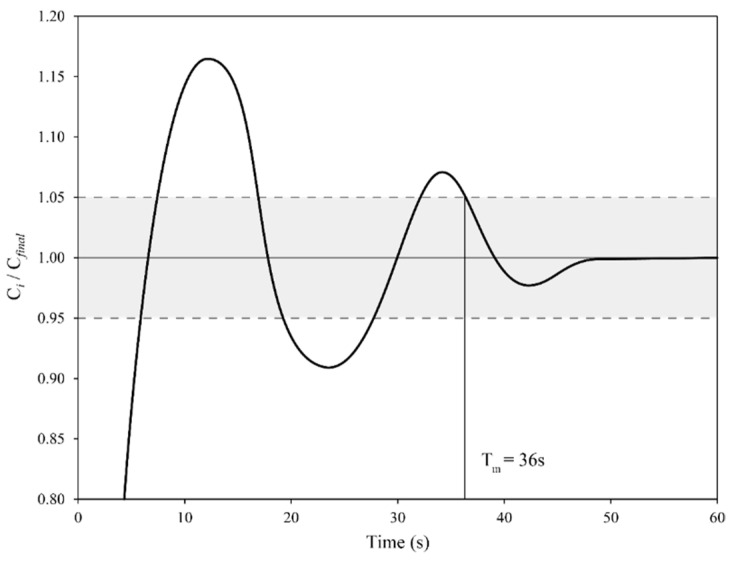
Graphical example of how mixing time can be calculated given the ratio C_i_/C_final_ over time.

**Figure 2 materials-14-04806-f002:**
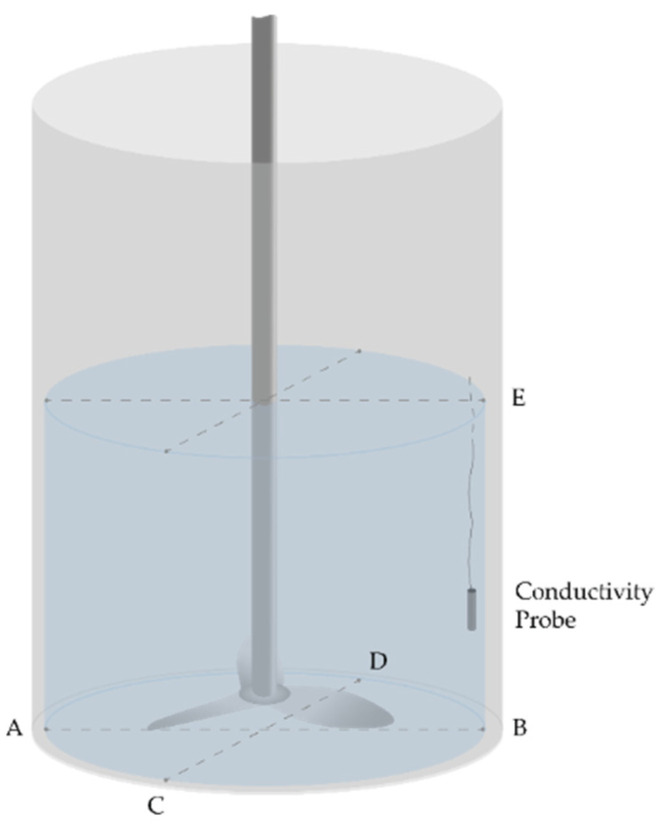
Schematic representing the mixing time trial setup. The probe positions are classified by the letters A, B, C, D and E.

**Figure 3 materials-14-04806-f003:**
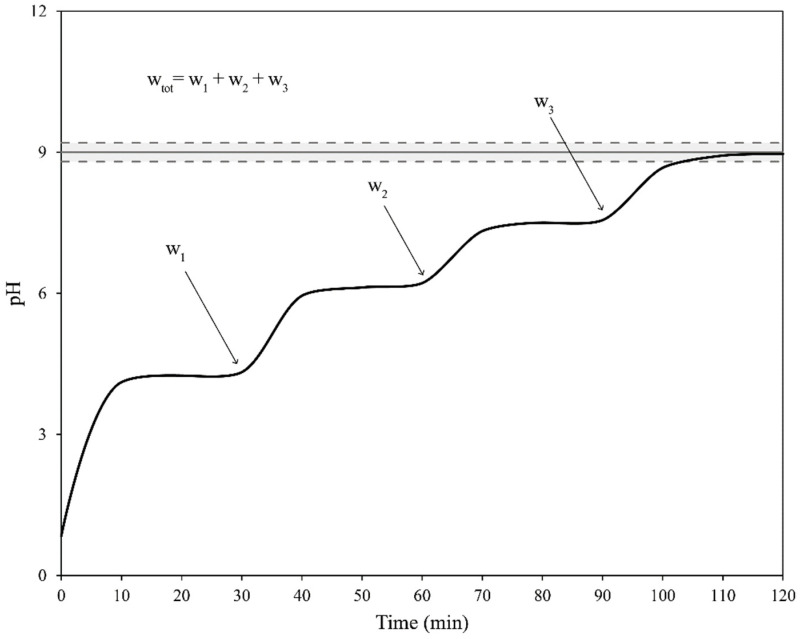
A schematic representation of a stepwise dosing trial with three additions of reactant.

**Figure 4 materials-14-04806-f004:**
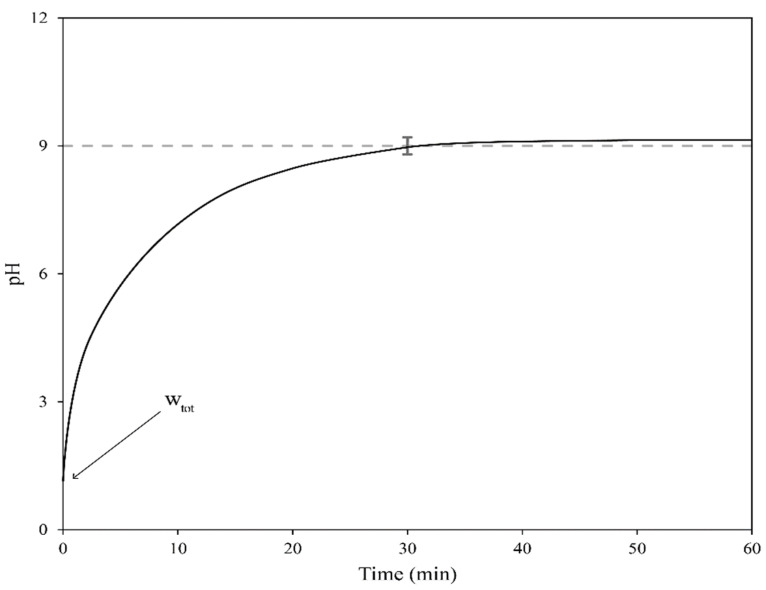
A schematic representation of a single-step dosing trial.

**Figure 5 materials-14-04806-f005:**
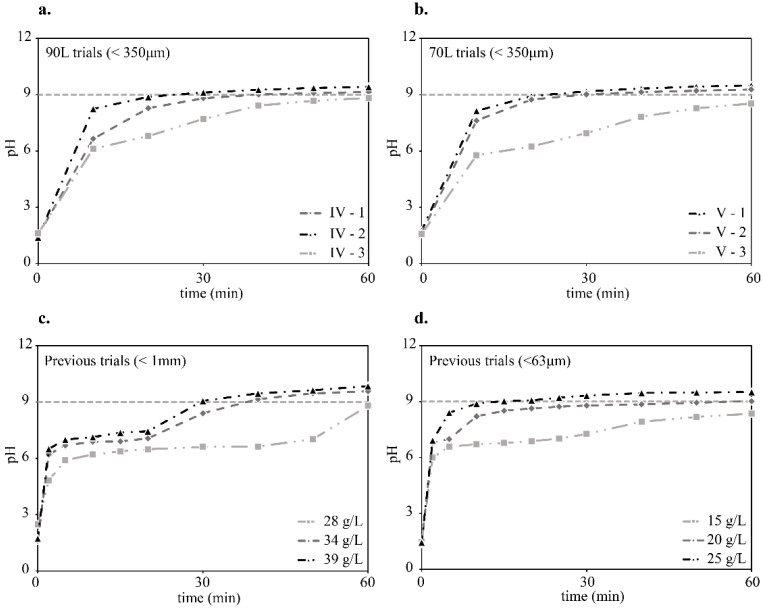
pH buffering obtained with the single-step methodology for volumes of wastewaters of 90 L (**a**) and 70 L (**b**). The results of previous trials conducted on 1 L, with the same slag type and the same company’s industrial wastewaters, are also presented. Trials with <1 mm powders are shown in (**c**) [9], while the ones with <63 µm powders are in (**d**) [10]. Adapted with permission from ref. [10]. Copyright 2019. Proceedings of the 6th International Slag Valorisation Symposium.

**Figure 6 materials-14-04806-f006:**
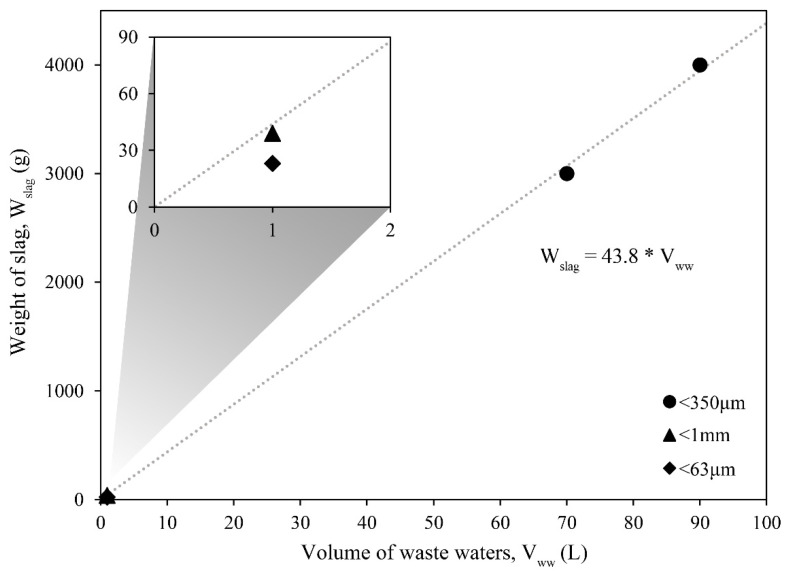
Linear regression of the 90 L and 70 L trials. Precedent trials have been added for comparison [9,10].

**Table 1 materials-14-04806-t001:** Summary of the mixing time trials classified by the rotational speed, volume of water and probe positions.

Trial No.	Volume (L)	Probe Positions	Speed (rpm)
1–3	90	A & B	225
4–6	80	A & B	225
7–9	70	A & B	225
10–12	80	A & B	200
13–15	70	A & B	175
16–18	90	A & B	175
19–21	90	C & D	225
22–24	80	C & D	200
25–27	70	C & D	175
28	90	A & E	225
29	80	A & E	225
30	70	A & E	225

**Table 2 materials-14-04806-t002:** Summary of stepwise and single-step dosing trials classified by volume, rotational speed and number of replications performed.

Type	Volume (L)	Speed (rpm)	Replications
Stepwise	90	175	2
Stepwise	70	175	1
Single-step	90	175	3
Single-step	70	175	3

**Table 3 materials-14-04806-t003:** Mixing time trials with probe positions in A & B.

Trials #	Volume (L)	Probe Positions	Speed (rpm)	T_m_ (s)	Average (s)
1	90	A & B	225	11	
2	90	A & B	225	10	
3	90	A & B	225	10	10.3
4	80	A & B	225	10	
5	80	A & B	225	9	
6	80	A & B	225	10	9.7
7	70	A & B	225	9	
8	70	A & B	225	10	
9	70	A & B	225	9	9.3
10	80	A & B	200	10	
11	80	A & B	200	12	
12	80	A & B	200	9	10.3
13	70	A & B	175	11	
14	70	A & B	175	10	
15	70	A & B	175	10	10.3
16	90	A & B	175	10	
17	90	A & B	175	11	
18	90	A & B	175	10	10.3

**Table 4 materials-14-04806-t004:** Mixing time trials with probe positions C & D and A & E.

Trials #	Volume (L)	Probe Positions	Speed (rpm)	T_m_ (s)	Average (s)
19	90	C & D	225	10	
20	90	C & D	225	9	
21	90	C & D	225	9	9.3
22	80	C & D	200	9	
23	80	C & D	200	9	
24	80	C & D	200	10	9.3
25	70	C & D	175	9	
26	70	C & D	175	10	
27	70	C & D	175	10	9.6
28	90	A & E	225	10	
29	80	A & E	225	8	
30	70	A & E	225	7	

**Table 5 materials-14-04806-t005:** Stepwise dosing trials liters, rpm, total mass of added slag, and final pH obtained.

Trial No.	Volume (L)	Speed (rpm)	m (kg)	Final pH
I	90	175	3	8.8
II	90	175	3	8.9
III	70	175	2	8.8

## Data Availability

No new data were created or analyzed in this study. Data sharing is not applicable to this article.

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
