# Peer review of "A Study of Treatment of Industrial Acidic Wastewaters with Stainless Steel Slags Using Pilot Trials"

_materials, 2021, doi:10.3390/ma14174806_

Round 1

Reviewer 1 Report

The article “A study of neutralization of industrial acidic waste waters with 2 stainless-steel slags using pilot trials” is discussing parameters for upscaling the neutralization of industrial acidic waste waters with stainless-steel slags. Results are compared with previous laboratory experiments using the same methodology.

There are still a series of questions and suggestions to be considered and answered before publication of this manuscript.

  1. The manuscript contains abbreviations which are not always explained when first introduced. E.g. AOD slag.
  2. Neutralization organic acidic waters is not the same as buffering acidic waters. Both terms are mixed up (conflict between title, abstract and manuscript). This should be cleared out as there is an essential difference between both descriptions. Please comment.

Neutralisation/neutralizing: lines 9, 14, 17, 335 and 432 versus buffering/buffer: lines 77, 89, 92, 98, 207, 213, 344, 345 353, 382, 396, 414, 416, 418, 425, 436, 453 and 456

  1. It is not clear if authors have performed a DOE performed starting experiments. Especially to see which parameters are dependent or independent and to reduce significantly the number of experiments. Please comment.
  2. It is not clear from Figure 2 how many point were selected to test different probe positions (8 in the manuscript, 5 in Fig.2). Please comment.
  3. Is there any thermal effect during mixing the slag in the acidic waters? Please comment.
  4. How pH measurement were performed since it is extremely difficult to obtain the pH at levels of 0.01? Which precautions have been taken to overcome experimental error when obtaining pH at such precision and accuracy (CO2 diffusion, temperature)? What is the experimental error?
  5. Line 364: A pH of 9 ±0.1 makes no sense. It should be 9.0±0.1? What is the reason that precision is now limited to 0.1? Please comment.
  6. It is not clear which experimental points are used for linear regression? Why 80 L is not considered? Please comment.
  7. The effect of the particle size is not clear: smaller (< 63 µm) and larger particle size (< 1 mm) with respect to particle size < 350µm as the latter is used for large volumes while the smaller ones only for 1 L. Please comment.
  8. ref. 13 is incomplete (no journal information).
  9. Line 231: “smaller than” instead of “lower than 350µm”

Author Response

Thank you very much for your comments, I found them very on point and precise and I worked towards their implementation. You can a find a more detailed explanation in the attached file. 

Best Regards

Mattia De Colle

Reviewer 2 Report

The paper is devoted to study of neutralization of industrial acidic waste waters with stainless-steel slags using pilot trials. The manuscript is well written and structured. I recommend this paper for minor revision.

I would suggest some improvements to Figure 5. The numbers are hardly readable due to a very small font size. 

Explain the ratio of 44 g/l slag to neutralize acidic waters (high consumption).

What are the expected volumes of acidic waste waters for neutralization?

How to utilize the generating by-product? What are the ways of slag disposal after neutralization?

Author Response

(The authors gave the same response as above.)

Reviewer 3 Report

Line/part

Reviewer’s comments

Substantive remarks

Materials and Methods

1. Has the research focused on the problem of the presence of heavy metals in slags, as mentioned by the authors in the Introduction (line 49-50)? Is there a risk that these metals end up in the treated wastewater?

2. As noted by the authors (lines 336-337), the slag does not dissolve completely in the wastewater, but remains in the form of a suspension. Is how much of the slag remains determined? Is it sinking to the bottom? What was the slag management after the process? Its amount is relatively large, so the use of the process on an industrial scale will leave the problem of final waste management.

Technical notes

56

Expand the AOD abbreviation

Tables 3 and 4

Provide the unit of Tm and Average

502-503

Complete the citation of the article: Ind. Eng. Chem. Res. 2007, 46, 14, 5032–5042. https://doi.org/10.1021/ie0613265

Author Response

(The authors gave the same response as above.)

Round 2

Reviewer 1 Report

I have read the revised version of the article “A study of neutralization of industrial acidic waste waters with 2 stainless-steel slags using pilot trials”.

Authors have made considerable efforts to answer questions and comments of reviewer in a detailed way and have implemented changes in the manuscript where needed.  

Therefore I support and recommend publication of the revised manuscript.  

Minor comment.

p.15 line 544: Moreover, the composition of the solid residues after the pH  buffering is highly depending on the treated acid composition …”